# On the difference between building and extracting patterns: a causal analysis of deep generative models.

## Abstract

Generative models are important tools to capture and investigate the properties of complex empirical data. Recent developments such as Generative Adversarial Networks (GANs) and Variational Auto-Encoders (VAEs) use two very similar, but *reverse*, deep convolutional architectures, one to generate and one to extract information from data. Does learning the parameters of both architectures obey the same rules? We exploit the causality principle of independence of mechanisms to quantify how the weights of successive layers adapt to each other. Using the recently introduced Spectral Independence Criterion, we quantify the dependencies between the kernels of successive convolutional layers and show that those are more independent for the generative process than for information extraction, in line with results from the field of causal inference. In addition, our experiments on generation of human faces suggest that more independence between successive layers of generators results in improved performance of these architectures.

## 1 Introduction

Deep generative models have proven powerful in learning to design realistic images in a variety of complex domains (handwritten digits, human faces, interior scenes). In particular, two approaches have recently emerged: Generative Adversarial Networks (GANs) (Goodfellow et al., 2014), which train an image generator by having it fool a discriminator that should tell apart real from artificially generated images; and Variational Autoencoders (VAEs) (Kingma & Welling, 2013; Rezende et al., 2014) that learn both a mapping from latent variables to the data (the decoder) and the converse mapping from the data to the latent variables (the encoder), such that correspondences between latent variables and data features can be easily investigated. Although these architecture have been lately the subject of extensive investigations, understanding why and how they work, and how they can be improved, remains elusive.

An interesting feature of GANs and VAEs is that they both involve the learning of two deep sub-networks. These sub-networks have a "mirrored" architecture, as they both consist in a hierarchy of convolutional layers, but information flows in opposite ways: generators and decoders map latent variables to the data space, while discriminators and encoders extract information from the same space. Interestingly, this difference could be framed in a causal perspective, with information flowing in the causal direction in the case of generators (from the putative causes of variations in the observed data), while extracting high level properties from the observations (with encoders or discriminators) would operate in the anti-causal direction.

Generative models in machine learning are usually not required to be causal, as modeling the data distribution is considered to be the goal to achieve. However, the idea that a generative model able to capture the causal structure of the data by disentangling the contribution of independent factors should perform better has been suggested in the literature (Bengio et al., 2013a; Mathieu et al., 2016) and evidence supports that this can help the learning procedure (Bengio et al., 2013b). Although many approaches have been implemented on specific examples, principles and automated ways of learning disentangled representations from data remains largely an open problem both for learning representations (targeting supervised learning applications) and for fitting good generative models. GANs have for example recently been a subject of intensive work in this direction, leading

to algorithms disentangling high level properties of the data such as InfoGans (Chen et al., 2016) or conditional GANs (Mirza & Osindero, 2014). However such models require supervision (e.g. feeding digit labels as additional inputs) to disentangle factors of interest. Unsupervised learning of disentangled representations has been addressed in various frameworks including Restricted Boltzmann Machines (Desjardins et al., 2012), tensor analyzers (Tang et al., 2013) and Lie groups (Cohen & Welling, 2014). A recent attempt to address unsupervised learning in VAEs is $\beta$-VAE (Higgins et al., 2017) which introduces and adjustable parameter $\beta$ in the VAE objective to strengthen the data compression constraint with respect to reconstruction error.

While the above approaches envision the disentangling of representations as finding subsets of latent variables that relate to different properties of the generated data, parameters of the network can be also considered as factors affecting the generated data. Ideally, to ensure modularity of deep generative models, different layers should encode different aspects of the data in their parameters. Intuitively for deep convolutional networks, different layers should encode image features at different scales. The idea that successive layers can be used as modules encoding different levels of details has for example been exploited to build high-resolution generative models by training iteratively a GAN with an increasing number of layers (Karras et al., 2017). Enforcing modularity of trained neural architecture may not only allow to adapt them to the task at hand with minimum additional training, but also to better understand the structure and function of these highly complex black-box systems. However, to the best of our knowledge, assessing how independent (or disentangled) are the properties encoded by the weights distributed across the structure of deep networks has not been addressed quantitatively in the literature.

We propose that the coupling between high dimensional parameters can be quantified and exploited in a causal framework to infer whether the layered architecture disentangles different aspects of the data. This hypothesis relies on recent work exploiting the postulate of Independence of Cause and Mechanism stating that Nature chooses independently the properties of a cause and those of the mechanism that generate effects from the cause (Janzing & Schölkopf, 2010; Lemeire & Janzing, 2012). Several methods relying on this principle have been proposed in the literature in association to different model classes (Janzing et al., 2010; Zscheischler et al., 2011; Daniusis et al., 2010; Janzing et al., 2012; Shajarisales et al., 2015; Sgouritsa et al., 2015; Schölkopf et al., 2012). Among these methods, the Spectral Independence Criterion (SIC) (Shajarisales et al., 2015) can be used in the context of linear dynamical systems, which involve a convolution mechanism.

In this paper, we show how SIC can be adapted to investigate the coupling between the parameters of successive convolutional layers. Empirical investigation shows that SIC is approximately valid between successive layers of generative models, and suggests SIC violations indicate deficiencies in the learning algorithm or the architecture of the network. Interestingly, and in line with theoretical predictions (Shajarisales et al., 2015), SIC tends to be more satisfied for generative sub-networks (mapping latent variables to data), than for the part of the system that map the anti-causal direction (data to latent variables). In addition, comparison of different generative models indicates that more independence between layers is associated to better performance of the model. Overall, our study suggests that quantifying Independence of Mechanisms in deep architecture can help analyze and design better generative models.

## 2 BACKGROUND

### 2.1 WHY A CAUSAL PERSPECTIVE?

An insightful description of causal models is based on Structural Equations (SEs) of the form

$$Y := f(X_1, X_2, \cdots, X_N, \epsilon)$$

The right hand side variable in this equation may or may not be random, and the additional independent noise term $\epsilon$, representing exogenous effects (originating from outside the system under consideration), may be absent. The ":=" symbol indicates the asymmetry of this expression and signifies that the left-hand-side variable is computed from the right-hand-side expression. This expression thus stays valid if something selectively changes on the right hand side variables, for example if the value of $X_1$ is externally forced to stay constant (hard intervention) or if the shape of $f$ or of an input distribution changes (soft intervention). These properties account for the robustness or invariance that is expected from causal models with respect to purely probalistic ones. Assume now $X_1$ itself

is determined by other variables according to

$$X_1 := g(U_1, U_2)$$

then the resulting structural causal model summarized by this system of equations also implies a modularity assumption: one can intervene on the second equation while the first one stays valid, baring the changes in the distribution of $X_1$ entailed by the intervention.

This structural equation framework describes well what would be expected from a robust generative model. Assume the model generates faces, one would like to be able to intervene on e.g. the pose without changing the rest of the top-level parameters (e.g. hair color), and still be able to observe a realistic output. One can also expect intervening on specific operations performed at intermediate levels by keeping the output qualitatively of the same nature. For example, we can imagine that by slightly modifying the mapping that positions the eyes with respect to the nose on a face, one may generate different heads that do not match exactly the standards of the dataset used for training, but are human-like. What we would not want is instead to see artifacts emerging all over the generated image, or edges being blurred.

## 2.2 INDEPENDENCE OF CAUSE AND MECHANISM (ICM)

Assume we have two variables $X$ and $Y$, possibly multidimensional and neither necessarily belonging to a vector space, nor necessary random. Assume the data generating mechanism obeys the following structural equation:

$$Y := m(X) \,,$$

with $m$ the mechanism, $X$ the cause and $Y$ the effect. We rely on the postulate that properties of $X$ and $m$ are "independent" in a sense that can be formalized in various ways (Peters et al., 2017). Broadly construed, ICM states that $m$ and $X$ do not carry information about each other, which can be expressed mathematically in a very general setting using algorithmic information Janzing & Schölkopf (2010). Based on this notion of independence, causal inference methods address the problem of identifying cause from effect, when both directions of causation ($X \to Y$ and $Y \to X$) are a priori plausible. Interestingly, there are several settings for which it is possible to derive an application specific quantification of ICM, such that if it is valid for the true causal direction (say $X \to Y$), the converse independence assumption is very likely violated (with high probability) for the anti-causal model

$$X := m^{-1}(Y)$$

(independence is then evaluated between $m^{-1}$ and $Y$). As a consequence, the true causal direction can be identified by picking the direction for which ICM is the most likely satisfied (see for example Janzing et al. (2010)).

## 2.3 SPECTRAL INDEPENDENCE

Shajarisales et al. (2015) introduce a specific formalization of ICM in the context of time series that will be suited to the study of convolutional layers in deep neural network. This relies on analyzing signals or images in the Fourier domain.

### 2.3.1 BACKGROUND ON DISCRETE SIGNALS AND IMAGES

The Discrete-time Fourier Transform (DTFT) of a sequence $a = \{a[k], k \in \mathbb{Z}\}$ is defined as

$$\widehat{\mathrm{a}}(\nu) = \sum_{k \in \mathbb{Z}} a[k] e^{-\mathbf{i} 2 \pi \nu k}, \ \nu \in \mathbb{R} \,.$$

Note that the DTFT of such sequence is a continuous 1-periodic function of the normalized frequency $\nu$. By Parseval's theorem, the energy (sum of squared coefficients) of the sequence can be expressed in the Fourier domain by $\|a\|_2^2 = \int_{-1/2}^{1/2} |\widehat{\mathrm{a}}(\nu)|^2 d\nu$. The Fourier transform can be easily generalized to 2D signals of the form $\{b[k, l], (k, l) \in \mathbb{Z}^2\}$, leading to a 2D function, 1-periodic with respect to both arguments

$$\widehat{\mathrm{b}}(u, v) = \sum_{k \in \mathbb{Z}, l \in \mathbb{Z}} b[k, l] e^{-\mathbf{i} 2 \pi (uk + vl)}, \ (u, v) \in \mathbb{R}^2 \,.$$

### 2.3.2 SIC POSTULATE

Assume now that our cause-effect pair $(X, Y)$ is a weakly stationary time series. This implies that the power of these signals can be decomposed in the frequency domain using their Power Spectral Densities (PSD) $S_x(\nu)$ and $S_y(\nu)$. If $Y$ results from the filtering of $X$ with convolution kernel $h$

$$Y := \left\{ \sum_{\tau \in \mathbb{Z}} h_\tau X_{t-\tau} \right\} = h * X \, . \tag{1}$$

then PSDs are related by the formula $S_y(\nu) = |\widehat{h}(\nu)|^2 S_x(\nu)$ for all frequencies $\nu$. The Spectral Independence Postulate consists then in assuming that the power amplification of the filter at each frequency $|\widehat{h}(\nu)|^2$ does not adapt to the input power spectrum $S_x(\nu)$, i.e. the filter will not tend to selectively amplify or attenuate the frequencies with particularly large or low power. This can be formalized by stating that the total output power (integral of the PSD) factorized into the product of input power and the energy of the filter, leading to the criterion (Shajarisales et al., 2015):

**Postulate 1** (Spectral Independence Criterion (SIC)). *Let $S_x$ be the Power Spectral Density (PSD) of a cause $X$ and $h$ the impulse response of the causal system of (1), then*

$$\int_{-1/2}^{1/2} S_x(\nu) |\widehat{h}(\nu)|^2 d\nu = \int_{-1/2}^{1/2} S_x(\nu) d\nu \cdot \int_{-1/2}^{1/2} |\widehat{h}(\nu)|^2 d\nu \, , \tag{2}$$

*holds approximately.*

We can define a scale invariant quantity $\rho_{X \to Y}$ measuring the departure from this SIC assumption, i.e. the dependence between input power spectrum and frequency response of the filter: the Spectral Dependency Ratio (SDR) from $X$ to $Y$ is defined as

$$\rho_{X \to Y} := \frac{\langle S_x \cdot |\widehat{h}|^2 \rangle}{\langle S_x \rangle \langle |\widehat{h}|^2 \rangle} \, , \tag{3}$$

where $\langle . \rangle$ denotes the integral (and also the average) over the unit frequency interval. The values of this ratio can be interpreted as follows: $\rho_{X \to Y} \approx 1$ reflects spectral independence, while $\rho_{X \to Y} > 1$ reflects a *correlation* between the input power and the filter's frequency response $|\widehat{h}(\nu)|^2$ across frequencies (the filter selectively amplifies input frequency peaks of large power, leading to anomalously large output power), conversely $\rho_{X \to Y} < 1$ reflects *anticorrelation* between these quantities. We will use these terms to analyze our experimental results. In addition Shajarisales et al. (2015) also derived theoretical results showing that if $\rho_{X \to Y} \approx 1$ for an invertible causal system, then $\rho_{Y \to X} < 1$ in the anti-causal direction. These interpretations of SDR values are summarized in Fig. 3b and can be used to interpret experimental results.

## 3 INDEPENDENCE OF MECHANISMS IN DEEP NETWORKS

We now introduce our causal reasoning in the context of deep convolutional networks, where the output of successive layers are often interpreted as different levels of representation of an image, from detailed low level features to abstract concepts. We thus investigate whether a form of modularity between successive layers can be identified using the above framework.

### 3.1 STRIDED CONVOLUTIONAL UNITS AND INDEPENDENCE BETWEEN SCALES

DCGANs have successfully exploited the idea of convolutional networks to generate realistic images (Radford et al., 2015). While strided convolutional units are used as pattern detectors in deep convolutional classifiers, they obviously play a different role in generative models: they are pattern producers. We provide in Fig. 1 (left) a toy example to explain their potential ability to generate independent features at multiple scales. On the picture the activation of one pixel in one channel at the top layer (top left) may encode the position of the pair of eyes in the image. After convolution by a first kernel, activations of a downstream channel in the second layer indicate the location of each eye. Finally, a kernel encoding the shape of the eye convolves this input to generate an image that combines the three types of information, distributed over three different spatial scales.

What we mean by assuming independence of the features encoded at different scales can be phrased as follows: there should be typically no strong relationship between the shape of patterns encoding

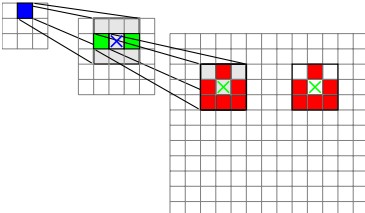 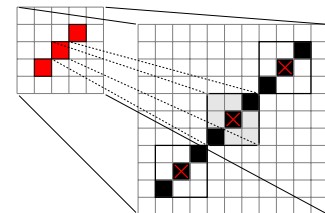

Figure 1: Left: schematic composition of coarse and finer scale features using two convolution kernels in successive layers to form the eyes of a cartoon face. Right: Example of violation of independence of mechanisms between two successive layers. In both cases crosses indicate center of patches (in light grey) affected by the activation of pixel in the previous layer.

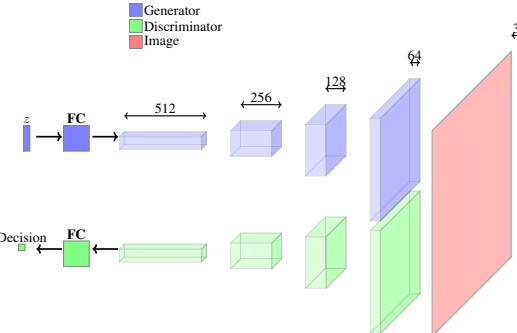

Figure 2: Architeture of the pretrained DCGAN generator used in our experiments. FC indicates a fully connected layer, $z$ is a 100-dimensional isotropic Gaussian vector, horizontal dimensions indicate the number of channels of each layer. The output image size is 64 by 64 pixels and these dimensions drop by a factor 2 from layer to layer.

a given object at successive scales. Although counter examples may be found, we postulate that a form of independence may hold approximately for a good generative model of naturalistic images that possess a large number of features at multiple scales. A case for which this assumption is violated due to limitations of the deep network architecture is given in Fig. 1, where a long edge in the image cannot be captured by a single convolution kernel (because of the kernel size limitation to 3 by 3 in this case). Hence, an identical kernel needs to be learned at an upper layer in order to control the precise alignment of activations in the bottom layer. Any misalignment between the orientation of the kernels would lead to a very different pattern, and witness that information in the two successive layers is entangled.

## 3.2 SIC BETWEEN SUCCESSIVE (DE)CONVOLUTIONAL UNITS

The SIC framework is well suited to the analysis of convolutional layers, since it assumes deterministic convolution mechanisms. However two differences appear with respect to the original assumptions: 1/ the striding adds spacing between input pixels in order to progressively increase the dimension and resolution of the image from one layer to the next, and 2/ there is a non-linearity between successive layers. Striding can be easily modeled as it amounts to upsampling the input image before convolution (see illustration Fig. 1). We denote $.^{\uparrow s}$ the upsampling operation with integer factor[1] $s$ that turns the 2D tensor $x$ into

$$x^{\uparrow s}[k,l] = \begin{cases} x[k/s, l/s], & k \text{ and } l \text{ multiple of s} \\ 0 & \text{otherwise.} \end{cases}$$

Interestingly, the definition implies that striding amounts to a compression of the normalized frequency axis in the Fourier domain with $\widehat{x^{\uparrow s}}(u,v) = \hat{x}(su, sv)$.

Next, the non-linear activation between successive layers is more challenging to take into account. We will thus make the simplifying assumption that rectified linear Units are used (ReLU), such that

---

[1] $s$ is the inverse of the stride parameter, which is fractional in that case

a pixel is either linearly dependent on small variations of its synaptic inputs, or not active at all. We then follow the idea that ReLU activations may be used as a switch that controls the flow of relevant information in the network (Tsai et al., 2016; Choi & Kim, 2017). Hence, for convolution kernels in successive layers that encode different aspects of the same object, we make the assumption that their corresponding outputs will be frequently coactivated, such that the linearity assumption may hold. If pairs of kernels are not frequently coactivated, we postulate that they do not encode the same object and it is thus unlikely that their weights are related.

Let us now write down the mathematical expression for the application of two successive deconvolutional units to a given input tensor using the above linearity assumptions. Let $z$ be the 2D activation map of a given channel in the first layer. It projects to the activation map $x$ of a channel in the second layer through convolution kernel $g$. Finally $x$ projects to the activation $y$ in the third layer through kernel $h$ (see Fig. 3c for an illustration). Successive upsamplings and convolutions lead to the expression

$$y = h * x^{\uparrow s} = h * (g * z^{\uparrow s})^{\uparrow s},$$

which corresponds to

$$\widehat{y}(u,v) = \widehat{h}(u,v)\widehat{x}(su,sv) = \widehat{h}(u,v)\widehat{g}(su,sv)\widehat{z}(s^2u,s^2.v),$$

in the Fourier domain. In order to have a criterion independent from the incoming activations in $z$, we assume $z$ has no spatial structure (e.g. $z$ is sampled from a 2D white noise), such that its power is uniformly distributed across spatial frequencies (the PSD is approximately flat). The spatial properties of $x$ are thus entirely determined by $g$ and the spectral independence criterion of equation 2 becomes

$$\left\langle |\hat{g}(u,v).\hat{h}(su,sv)|^2 \right\rangle = \left\langle |\widehat{g}(u,v)|^2 \right\rangle \left\langle |\widehat{h}(su,sv)|^2 \right\rangle, \tag{4}$$

where angular brackets denote the double integral across spatial frequencies. We can then write down an updated version of the SDR of equation 3 corresponding to testing SIC between a cascade of two filters in successive layers as

$$\rho_{g \to h} = \frac{\left\langle |\widehat{g}(u,v)\widehat{h}(su,sv)|^2 \right\rangle}{\langle |\widehat{g}(u,v)|^2 \rangle \left\langle |\widehat{h}(u,v)|^2 \right\rangle},$$

which we will evaluate in the experiments.

# 4 EXPERIMENTS

We used a version of DCGANs pretrained on the CelebFaces Attributes Dataset (CelebA)[2]. The structures of the generator and discriminator are summarized in Fig. 2 and both include four successive convolutional layers. We also experimented with a plain VAEs[3] with a similar convolutional architecture with the following differences: the default number of channels of the bottom hidden layer is 32 (but we change it to 64 in the last subsection), and the dimension of the latent vector $z$ is 128. Each layer uses 4 pixels wide convolution kernels for the GAN, and 5 pixels wide for the VAE. In all cases, layers are enumerated following the direction of information flow. We will talk about coarser scales for layers that are towards the latent variables, and finer scales for the ones that are closer to the image.

## 4.1 SIC BETWEEN SUCCESSIVE DECONVOLUTIONAL UNITS

We first illustrate the previously introduced framework by showing in Fig. 3a an example of two convolution kernels between successive layers (1 and 2) of the GAN generator. It shows a slight anti-correlation between the absolute values of the Fourier transforms of both kernels, resulting I a SDR of .88. One can notice the effect of kernels $h$ (upper layer) and $g$ (lower layer) on the resulting convolved kernel in the Fourier domain: the kernel from the upper layer tends to modulate the fast variations of $g * h$ in the Fourier domain, while $g$ affects the 'slow' variations. This is a consequence of the design of the strided fractional convolution. We use this approach to characterize the amount

---

[2] http://mmlab.ie.cuhk.edu.hk/projects/CelebA.html
[3] https://github.com/yzwxx/vae-celebA

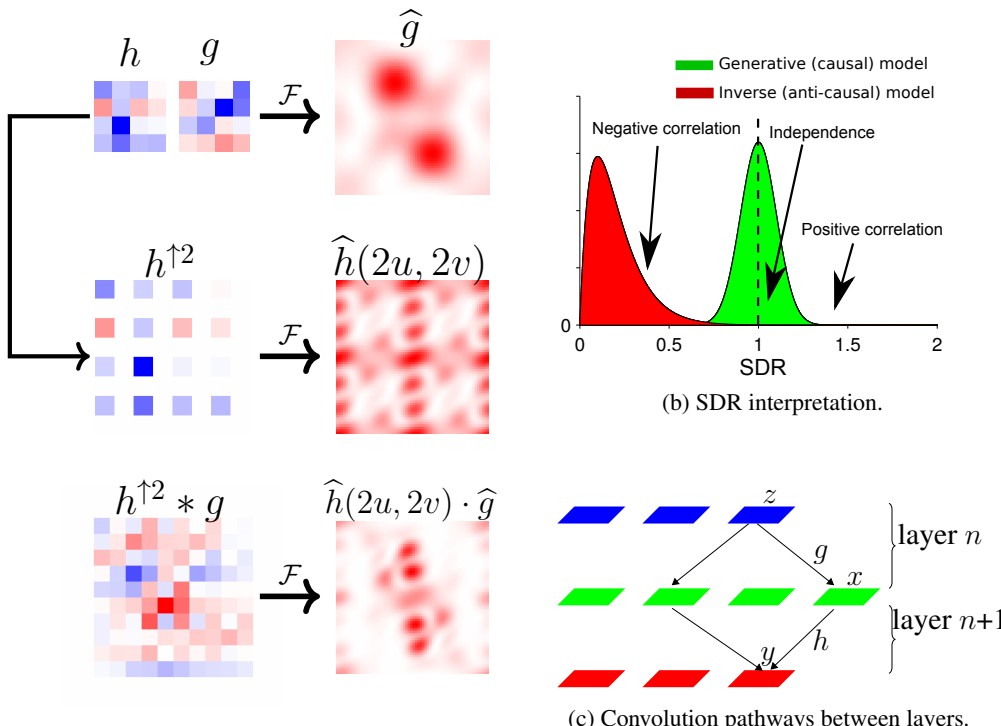

(a) Illustration of successive convolutions.

Figure 3: 3a Example convolution kernels and corresponding Fourier transforms (zero frequencies are located at the center of each picture), taken from layer 1 (for $h$) and 2 (for $g$) of the generator of a trained DCGAN. 3b Illustration of the meaning of SDR values. 3c Illustration of the multiple compositions of convolution kernels belonging to successive layers. The pathway depends on the considered input layer (blue), output layer (red) and intermediate layer (green) channels.

of dependency between successive layers by plotting the histogram of the SDR that we get for all possible combination of kernels belonging to each layer, i.e. all possible pathways between all input and output channels, as described in Fig. 3c. The result is shown in Fig. 4, witnessing a rather good concentration of the SDR around 1, which suggests independence of the convolution kernels between successive layers. It is interesting to compare these histograms to the values computed for the discriminator of the GAN, which implements convolutional layers of the same dimensions in reverse order. The result, also shown in Fig. 4, exhibits a broader distribution of the SDR, especially for layers encoding lower level image features. This is in line with the principle of independence of mechanism, as the discriminator is operating in the anticausal direction. However, the difference between the generator and discriminator is not strong, which may be due to the fact that the discriminator does not implement an inversion of the putative causal model, but only extract relevant latent information for discrimination. In order to check our method on a generative model including a network mapping back the input to their cause latent variables, we applied it to a trained VAE. The results presented in Fig. 5 show much sharper differences between generator (decoder) and encoder. The shape of the histograms are matching predictions from (Shajarisales et al., 2015) shown in Fig. 3b: with a mode around one for the distribution in the causal direction, while it remains below one in the anti-causal direction. The difference with GANs can be explained by the fact that VAEs are indeed performing an inversion of the generative model, leading to very small SDR values in the anticausal direction. We also note overall a much broader distribution of VAE SDRs in the causal direction (decoder) with respect to their GAN counterpart. Interestingly, the training of the VAE did not lead to values as satisfactory as what we obtained with the GAN. Examples generated from the VAE are shown in appendix (Fig. 11). This suggests that dependence between cause and mechanism may reflect a suboptimal performance of the generator.

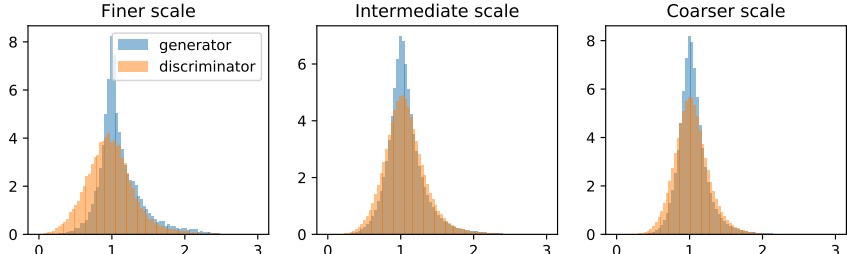

Figure 4: Superimposed histograms of spectral density ratios of a trained GAN generator and discriminator for layers at the same level of resolution (left to right from finer to coarser).

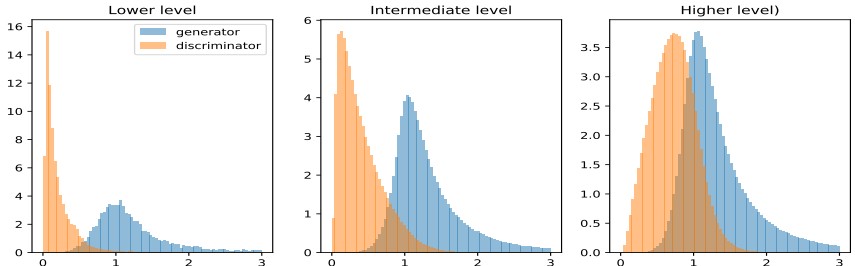

Figure 5: Superimposed histograms of spectral density ratios of trained VAE generator and discriminator for layers at the same level of resolution (left to right from finer to coarser).

## 4.2 REMOVING CORRELATED UNITS

We saw in the above histograms that while many successive convolutional unit had a SDR close to one, there are tails composed of units exhibiting either a ratio inferior to one (reflecting negative correlation between the Fourier transform of the corresponding kernels) or a ratio larger than one (reflecting positive correlation). Interestingly, if we superimpose the histograms (see Fig. 4) of the lower level layer of the generator and discriminator networks, we see that these tails are quite different between networks, showing more negative correlations for the discriminative networks, while the positive correlation tail of the generative networks remains rather strong. This suggests that negative and positive correlation are qualitatively different phenomena. In order to investigate the nature of filters exhibiting different signs of correlation, we selectively removed filters of the third layer of the generator (the last but one), based on the magnitude (above or below one) of the average SDR that each filter achieved when combined with any of the filters in the last layer. In order to check that our results were not influenced by filters with very small weights (that still can exhibit correlations), we zeroed the kernels contributing to the output layer with the smallest energy (averaged across output channels), while maintaining an acceptable quality of the output of the network (see Fig. 6 second column). This removed around half of the filters of the third layer. Then we removed additional filters exhibiting either large or small (anti-correlated) generic ratio, such that the same proportion of filters is removed (see Fig. 6 third and fourth column). It appears clearly from this result that filters exhibiting large positive or negative correlation do not play the same role in the network. From the quality of the generated images, filters from the third layer that are negatively correlated to filters from the fourth seem to allow correction of checkerboard artifacts, potentially generated as side effect of the fractional strided convolution mechanisms[4]. Despite a decrease in texture quality, the removal of such kernels does not distinctively affect an meaningful aspect of the generated images. Conversely, removing positively correlated filters lead to a disappearance of the color information in a majority of the regions of the image. This suggests that such filters do encode the color structure of images, and the introduced positive correlation between the filters from the third and fourth layer may result from the fact that uniform color patches corresponding to specific parts of the image (hair, skin) need a tight coordination of the filters at several scales of the image in order to produce large uniform areas with sharp border. As we observed more dependency betwen GAN layers at fine scales, we considered using dropout (Srivastava et al., 2014) in order to reduce

---

[4] https://distill.pub/2016/deconv-checkerboard/

| original | reduced | <0 correlation removed | >0 correlation removed |

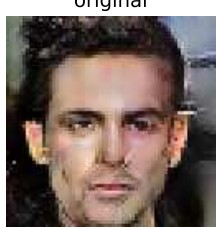 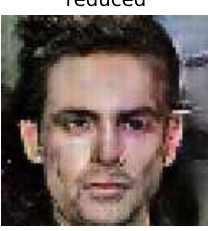 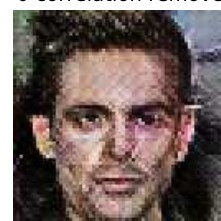 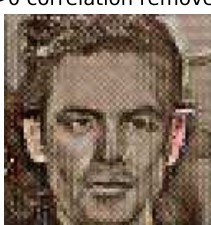

Figure 6: Random face generated by a pretrained DCGAN (left column). Second column: output of the same network when removing low energy filters from the third layer (reduction from 8,192 to 4,260). Third column: output of the same network when removing the filters leading to the lowest average SDR ($\rho < .9$, leading to 3,684 filters). Fourth column: same when removing the filters leading to the largest SDR ($\rho > 1.45$, leading to 3,660 filters). More examples are shown in appendix Fig. 10.

this dependency. Indeed, dropout has been introduced with the idea that it can prevent neurons from over-adapting to each other and thus regularize the network. The results shown in Fig. 12 witness on the contrary an increase of the dependencies (especially positive correlation) between these layers and exhibit a strongly deteriorated performance as shown in examples Fig. 13. We suggest that dropout limits the expressivity of the network by enforcing more redundancy in the convolutional filters, leading also to more dependency between them.

### 4.3 RELATIONSHIP BETWEEN PERFORMANCE AND SDR STATISTICS.

In order to further assess how much our SIC values relate to the performance of the generative model, we trained different versions of VAEs by acting on two parameters: we changed the number of channels in the last hidden layer, and weighted differently the least square reconstruction error term with respect to the Kullback-Leibler (KL) divergence term (reflecting data compression) in the VAE objective by applying a multiplicative factor $\beta$ to the KL divergence, in the same way as it is done for $\beta$-VAEs (Higgins et al., 2017). Example results on Fig. 7 show that while increasing the number of channels possibly provides a slight improvement, decreasing $\beta$ strongly increases the variety of faces that can be generated by the network. In particular, VAEs trained with a lower $\beta$ (right-hand-side), exhibit a broader variety of hair styles, face shapes and background colors. Interestingly, this improvement is reflected in the SDR statistics plotted in Fig. 8, showing more concentrated SDR values for better performing networks at the coarser level, which likely relates to the type of broad variations in face characteristics introduced by a decrease in $\beta$. Direct comparison of SDR distributions between the worst and best performing models is provided in Fig. 9 and confirms the above observations. Interestingly, we can also identify the SDR values at the finest level are a slightly more concentrated around one for the worst performing model. Although this seems to contradict our previous statement, a careful observation of the fine details of the generated pictures in Fig. 7 shows that the best performing model (bottom right) is far from perfect at this scale and contains more pixelation artifacts than the worst performing one (top left). While further improving the VAE training and architecture is beyond the scope of this paper, these results suggests that SDR statistics can be used as a diagnostic tool to guide the choice of parameters of the architecture and identify which module needs improvement.

## 5 DISCUSSION

In this work, we derived a measure of independence between the weights learned by convolutional layers of deep networks. The results suggest that generative models that map latent variables to data tend to have more independence between successive layers than discriminative or encoding networks. This is in line with theoretical predictions about independence of mechanisms for causal and anti-causal systems. In addition, our results suggest the dependency between successive layers relates to the bad performance of the trained generative models. Moreover, the SDR analysis also indicates which layers should be modified to improve the performance. Enforcing independence during training may thus help diagnose and improve generative models. Finally, we speculate that independence between successive layers, by favoring modularity of the network, may help build

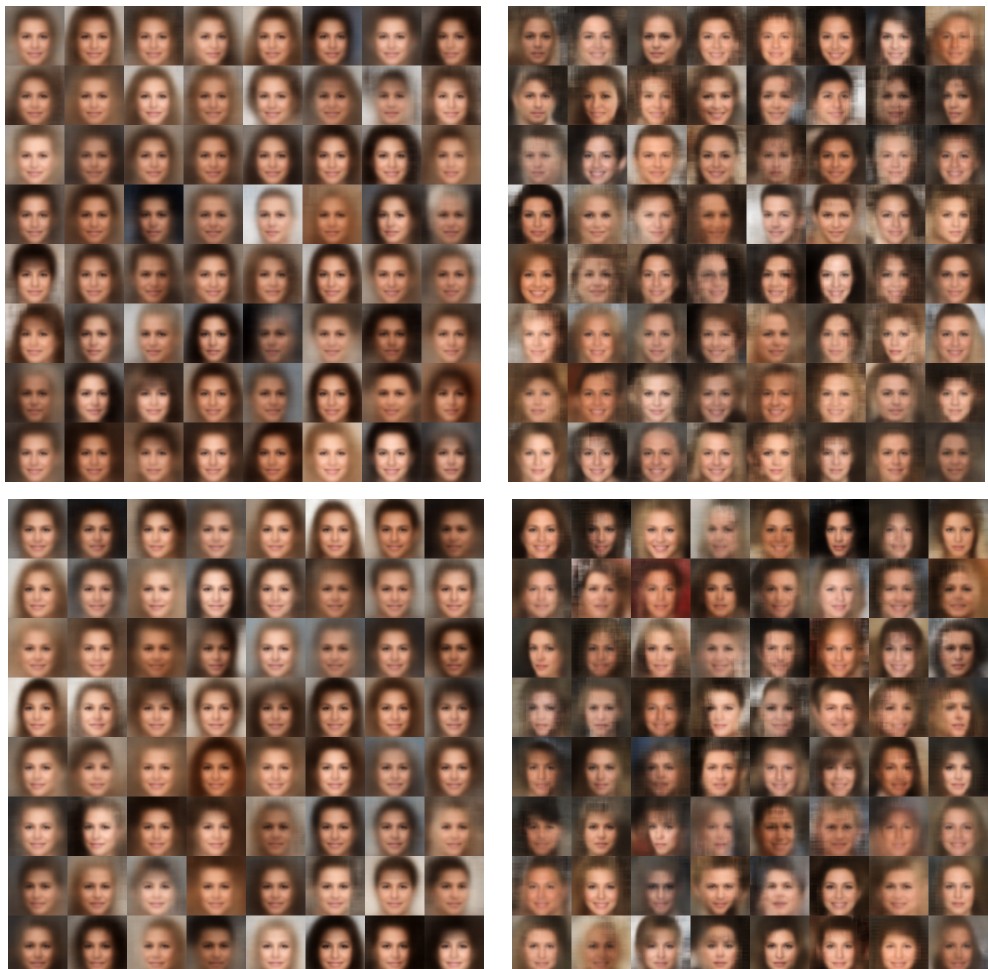

Figure 7: Example faces generated by VAEs with different combinations of network parameters: number channels in the last hidden layer and $\beta$ parameter. Top: 32 channels, bottom: 64 channels. Left: $\beta = 1$, right: $\beta = .1$.

architectures that can be easily adapted to new purposes. In particular, separation of spatial scales in such models may help build networks in which one can intervene on one scale without affecting others, with applications such as style transfer Gatys et al. (2015). One specific feature of our approach is that this quantitative measure of the network performance in not statistical and as such requires neither extensive sampling form the fitted generative distribution nor from real datasets to be computed: only the parameters of the model are used. This is in strong contrast with state-of-the-art approaches such as the Fréchet Inception Distance (FID) (Heusel et al., 2017), and makes the approach easy to apply to any neural network equipped with convolutional layers.

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

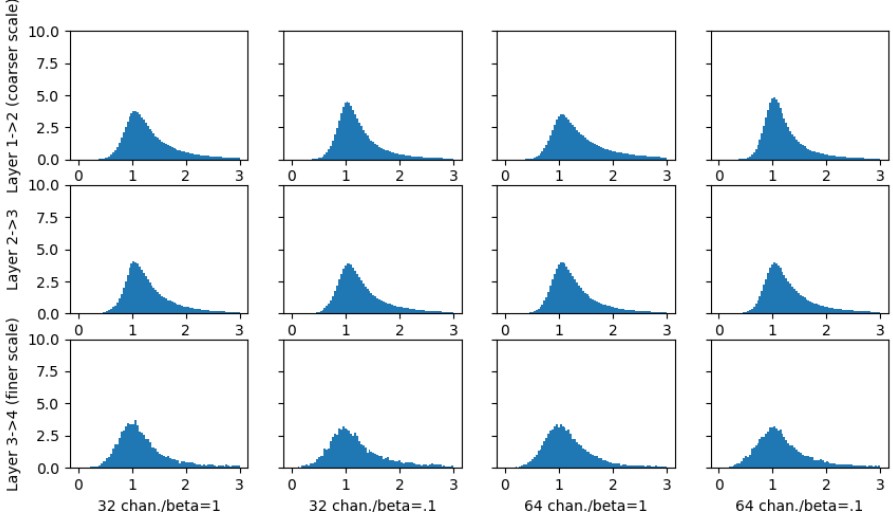

Figure 8: SDR encoder histograms for different choices of VAE parameters.

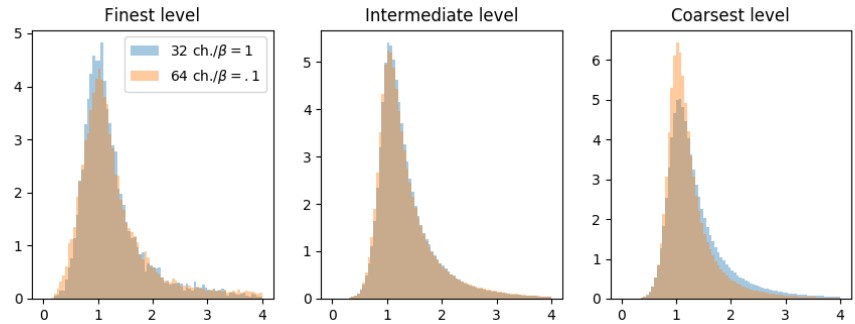

Figure 9: Comparison of SDR decoder histograms between worst and best performing VAEs (32 channels/$\beta = 1$ and 64 channels/$\beta = .1$, respectively).

T. Cohen and M. Welling. Learning the irreducible representations of commutative lie groups. In *International Conference on Machine Learning*, pp. 1755–1763, 2014.

P. Daniusis, D. Janzing, K. Mooij, J. Zscheischler, B. Steudel, K. Zhang, and B. Schölkopf. Inferring deterministic causal relations. In *UAI 2010*, 2010.

G. Desjardins, A. Courville, and Y. Bengio. Disentangling factors of variation via generative entangling. *arXiv preprint arXiv:1210.5474*, 2012.

L. A. Gatys, A. S. Ecker, and M. Bethge. A neural algorithm of artistic style. *arXiv preprint arXiv:1508.06576*, 2015.

I. Goodfellow, J. Pouget-Abadie, M. Mirza, B. Xu, D. Warde-Farley, S. Ozair, A. Courville, and Y. Bengio. Generative adversarial nets. In *Advances in neural information processing systems*, pp. 2672–2680, 2014.

M. Heusel, H. Ramsauer, T. Unterthiner, B. Nessler, G. Klambauer, and S. Hochreiter. Gans trained by a two time-scale update rule converge to a nash equilibrium. *arXiv preprint arXiv:1706.08500*, 2017.

I. Higgins, L. Matthey, A. Pal, C. Burgess, X. Glorot, M. Botvinick, S. Mohamed, and A. Lerchner. beta-vae: Learning basic visual concepts with a constrained variational framework. In *ICLR 2017*, 2017.

D. Janzing and B. Schölkopf. Causal inference using the algorithmic Markov condition. *Information Theory, IEEE Transactions on*, 56(10):5168–5194, 2010.

D. Janzing, P.O. Hoyer, and B. Schölkopf. Telling cause from effect based on high-dimensional observations. In *Proceedings of the 27th International Conference on Machine Learning (ICML-10)*, 2010.

D. Janzing, J. Mooij, K. Zhang, J. Lemeire, J. Zscheischler, P. Daniušis, B. Steudel, and B. Schölkopf. Information-geometric approach to inferring causal directions. *Artificial Intelligence*, 182–183:1–31, 2012.

T. Karras, T. Aila, S. Laine, and J. Lehtinen. Progressive growing of gans for improved quality, stability, and variation. *arXiv preprint arXiv:1710.10196*, 2017.

D. P. Kingma and M. Welling. Auto-encoding variational bayes. *arXiv preprint arXiv:1312.6114*, 2013.

J. Lemeire and D. Janzing. Replacing causal faithfulness with algorithmic independence of conditionals. *Minds and Machines*, pp. 1–23, 7 2012. doi: 10.1007/s11023-012-9283-1.

M. F. Mathieu, J. J. Zhao, A. Ramesh, P. Sprechmann, and Y. LeCun. Disentangling factors of variation in deep representation using adversarial training. In *Advances in Neural Information Processing Systems*, pp. 5041–5049, 2016.

M Mirza and S Osindero. Conditional generative adversarial nets. *arXiv preprint arXiv:1411.1784*, 2014.

J. Peters, D. Janzing, and B. Schölkopf. *Elements of Causal Inference – Foundations and Learning Algorithms*. MIT Press, 2017.

A. Radford, L. Metz, and S. Chintala. Unsupervised representation learning with deep convolutional generative adversarial networks. *arXiv preprint arXiv:1511.06434*, 2015.

D. J. Rezende, S. Mohamed, and D. Wierstra. Stochastic backpropagation and approximate inference in deep generative models. *arXiv preprint arXiv:1401.4082*, 2014.

B. Schölkopf, D. Janzing, J. Peters, E. Sgouritsa, K. Zhang, and J. Mooij. On causal and anticausal learning. In *ICML 2012*, 2012.

E. Sgouritsa, D. Janzing, P. Hennig, and B. Schölkopf. Inference of cause and effect with unsupervised inverse regression. In *ICML 2015*, 2015.

N. Shajarisales, D. Janzing, B. Schölkopf, and M. Besserve. Telling cause from effect in deterministic linear dynamical systems. In *ICML 2015*, 2015.

N. Srivastava, G. E. Hinton, A. Krizhevsky, I. Sutskever, and R. Salakhutdinov. Dropout: a simple way to prevent neural networks from overfitting. *Journal of machine learning research*, 15(1):1929–1958, 2014.

Y. Tang, R. Salakhutdinov, and G. Hinton. Tensor analyzers. In *International Conference on Machine Learning*, pp. 163–171, 2013.

C-Y. Tsai, A. M. Saxe, and D. Cox. Tensor switching networks. In *Advances in Neural Information Processing Systems*, pp. 2038–2046, 2016.

J. Zscheischler, D. Janzing, and K. Zhang. Testing whether linear equations are causal: A free probability theory approach. In *UAI 2011*, 2011.

APPENDIX

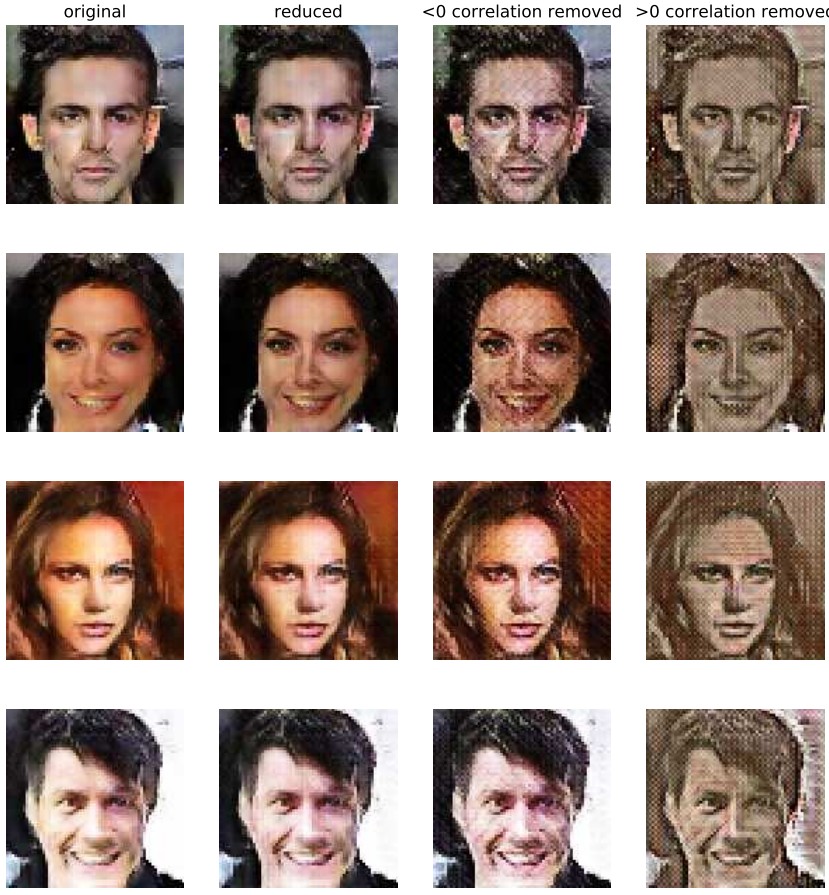

Figure 10: Example generated figures using a pretrained DCGAN (left column). Second column: the output of the same network when removing low energy filters from the third layer (reduction of the number of filters from 8,192 to 4,260). Third column: the output of the same network when removing the filters leading to the lowest average SDR ($\rho < .9$, leading to 3,684 filters). Fourth column: same when removing the filters leading to the largest average SDR ($\rho > 1.45$, leading to 3,660 filters).

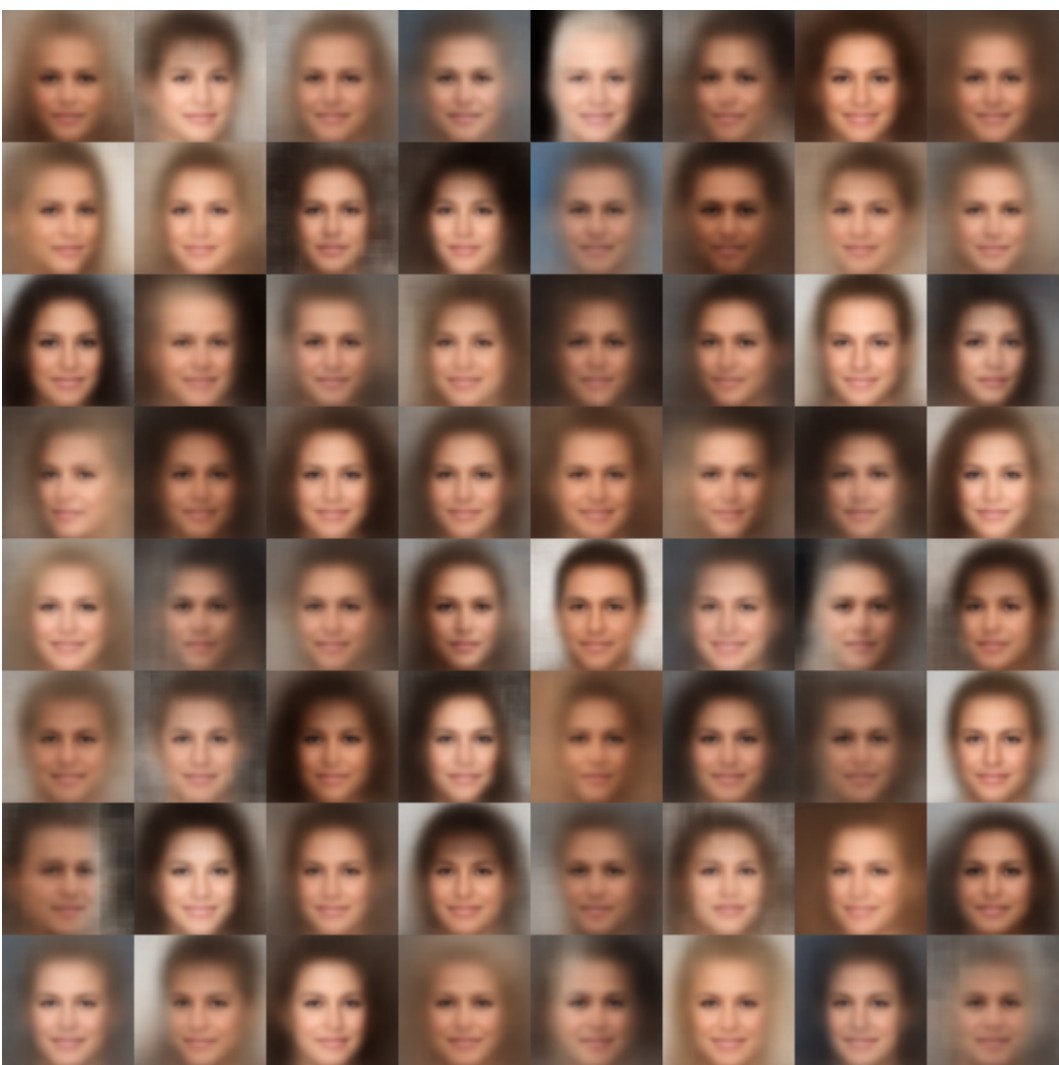

Figure 11: Random examples generated by a trained VAE.

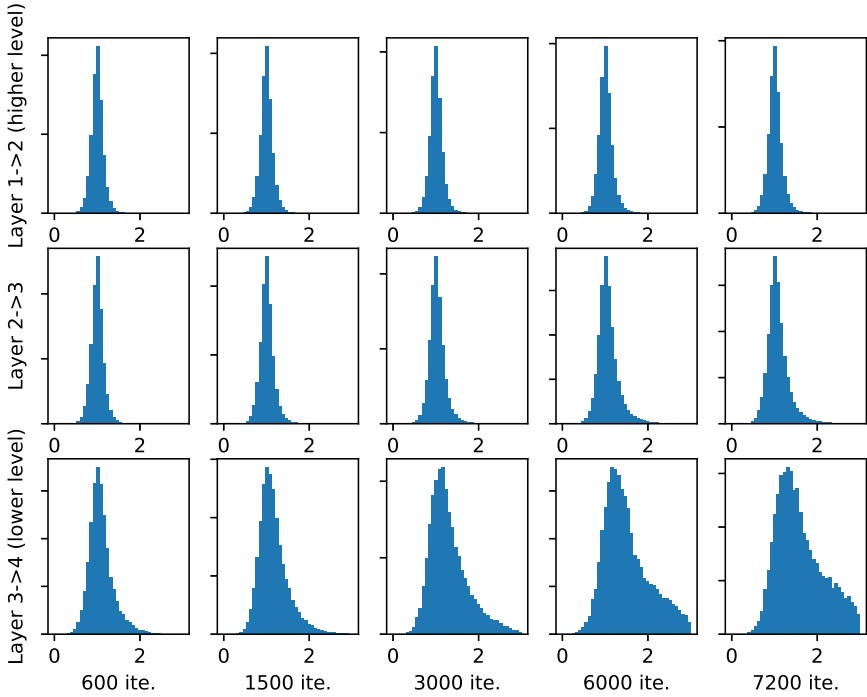

Figure 12: Evolution of the spectral density ratios between successive layers as a function of training iteration when dropout is used between the two finer scale layers of the generator.

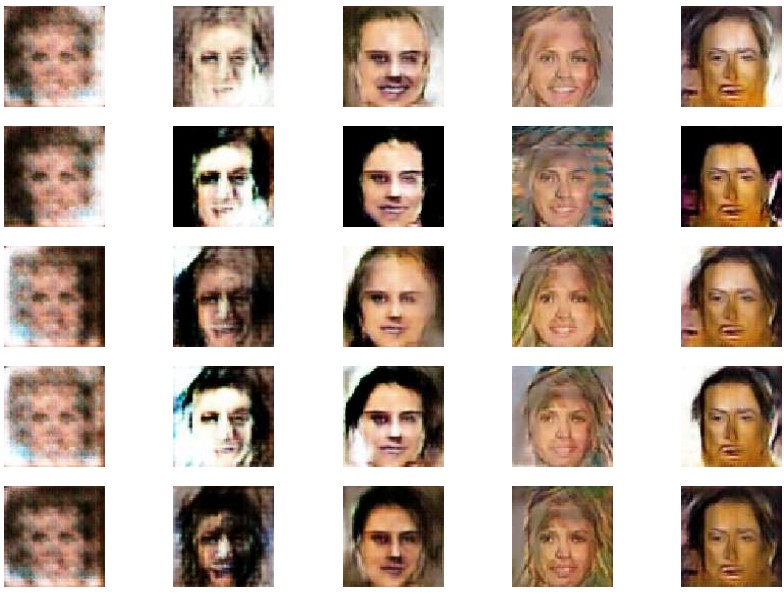

Figure 13: Evolution of generated examples (for a fixed latent input) as function of training iteration (same as Fig. 12) when dropout is used.

