# OpenReview forum: "On the difference between building and extracting patterns: a causal analysis of deep generative models."
_ICLR.cc/2018/Conference — Reject_

### Official Review · AnonReviewer2 · 2017-11-15
**Interesting metric, but no actionable steps or improvement results**

**Rating:** 2
**Confidence:** 4

**Review:**

This paper examines the nature of convolutional filters in the encoder and a decoder of a VAE, and a generator and a discriminator of a GAN. The authors treat the inputs (X) and outputs (Y) of each filter throughout each step of the convolving process as a time series, which allows them to do a Discrete Time Fourier Transform analysis of the resulting sequences. By comparing the power spectral density of the input and the output, they get a Spectral Dependency Ratio (SDR) ratio that characterises a filter as spectrally independent (neutral), correlating (amplifies certain frequencies), or anti-correlating (dampens frequencies). This analysis is performed in the context of the Independence of Cause and Mechanism (ICM) framework. The authors claim that their analysis demonstrates a different characterisation of the inference/discriminator and generative networks in VAE and GAN, whereby the former are anti-causal and the latter are causal in line with the ICM framework. They also claim that this analysis can be used to improve the performance of the models.

Pros:
-- SDR characterisation of the convolutional filters is interesting
-- The authors show that filters with different characteristics are responsible for different aspects of image modelling

Cons:
-- The authors do not actually demonstrate how their analysis can be used to improve VAEs or GANs
-- Their proposed SDR analysis does not actually find much difference between the generator and the discriminator of the GAN
-- The clarity of the writing could be improved (e.g. the discussion in section 3.1 seems inaccurate in the current form). Grammatical and spelling mistake are frequent. More background information could be helpful in section 2.2. All figures (but in particular Figure 3) need more informative captions
-- The authors talk a lot about disentangling in the introduction, but this does not seem to be followed up in the rest of the text. Furthermore, they are missing a reference to beta-VAE (Higgins et al, 2017) when discussing VAE-based approaches to disentangled factor learning


In summary, the paper is not ready for publication in its current form. The authors are advised to use the insights from their proposed SDR analysis to demonstrate quantifiable improvements the VAEs/GANs.

---

### Official Review · AnonReviewer1 · 2017-11-25
**an interesting analysis of deep generative models using causality**

**Rating:** 7
**Confidence:** 3

**Review:**

This work exploits the causality principle to quantify how the weights of successive layers adapt to each other.  Some interesting results are obtained, such as "enforcing more independence between successive layers of generators may lead to better performance and modularity of these architectures" . Generally, the result is interesting and the presentation is easy to follow. However, the proposed approach and the experiments are not convincible enough.  For example,  it is hard to obtain the conclusion "more independence lead to better performance" from the experimental results. Maybe more justifications are needed.

---

### Official Review · AnonReviewer3 · 2017-11-27
**As far as I could understand, interesting and potentially useful**

**Rating:** 7
**Confidence:** 2

**Review:**

The paper presents an application of a measure of dependence between the input power spectrum and the frequency response of a filter (Spectral Density Ratio from [Shajarisales et al 2015]) to cascades of two filters in successive layers of deep convolutional networks. The authors apply their newly defined measure to DCGANs and plain VAEs with ReLUs, and show that dependency between successive layers may lead to bad performance.

The paper proposed a possibly interesting approach, but I found it quite hard to follow, especially Section 4, which I thought was quite unstructured. Also Section 3 could be improved and simplified. It would be also good to add some more related work. I’m not an expert, but I assume there must be some similar idea in CNNs.

From my limited point of view, this seems like a sound, novel and potentially useful application of a interesting idea. If the writing was improved, I think the paper may have even more impact.

Smaller details: some spacing issues, some extra punctuation (pg 5 “. . Hence”), a typo (pg. 7 “training of the VAE did not lead to values as satisfactory AS what we obtained with the GAN”)

---

### Decision · Program_Chairs · 2018-01-29
**ICLR 2018 Conference Acceptance Decision**

**Decision:**

Reject

**Comment:**

Thank you for submitting you paper to ICLR. The paper presents an interesting analysis, but the utility of this analysis is questionable e.g. it is not clear how this might lead to improved VAEs/GANs. The authors did add an additional experimental result in their revised paper, but questions still remain. In light of this the significance of the paper is on the low side and it is therefore not ready for publication in ICLR without more work.